# Impact of early-onset persistent stunting on cognitive development at 5 years of age: Results from a multi-country cohort study

Md Ashraful Alam[1], Stephanie A. Richard[2], Shah Mohammad Fahim[1], Mustafa Mahfuz[1]*, Baitun Nahar[1], Subhasish Das[1], Binod Shrestha[3], Beena Koshy[4], Estomih Mduma[5], Jessica C. Seidman[2], Laura E. Murray-Kolb[6], Laura E. Caulfield[7], Tahmeed Ahmed[1]

1 icddr,b, Shaheed Tajuddin Ahmed Sarani, Mohakhali, Dhaka, Bangladesh, 2 Fogarty International Center/ National Institutes of Health, Bethesda, MD, United States of America, 3 Water Reed/AFRIMS Research Unit Nepal (WARUN), Kathmandu, Nepal, 4 Christian Medical College, Vellore, India, 5 Haydom Lutheran Hospital, Haydom, Tanzania, 6 The Pennsylvania State University, University Park, PA, United States of America, 7 The Johns Hopkins University, Baltimore, MD, United States of America

* mustafa@icddrb.org

## Abstract

### Background

Globally more than 150 million children under age 5 years were stunted in 2018, primarily in low- and middle-income countries (LMICs), and the impact of early-onset, persistent stunting has not been well explored. To explore the association between early-onset persistent stunting in children and cognitive development at 5 years of age, and to identify the factors associated with early-onset stunting.

### Methods and findings

Children from the MAL-ED cohort study were followed from birth to 5 years of age in six LMICs. The Wechsler Preschool Primary Scales of Intelligence (WPPSI) was used to assess cognitive abilities (fluid reasoning) at 5 years and was adapted for each culture. Stunting was categorized as early-onset persistent (first stunted at 1–6 months and persisting at 60 months), early-onset recovered (first stunted at 1–6 months and not stunted at 60 months), late-onset persistent (first stunted at 7–24 months and persisting at 60 months), late-onset recovered (first stunted at 7–24 months and not stunted at 60 months), and never (never stunted). Mixed effects linear models were used to estimate the relationship between stunting status and cognitive development. Children with early-onset persistent stunting had significantly lower cognitive scores (-2.10 (95% CI: -3.85, -0.35)) compared with those who were never stunted. Transferrin receptor (TfR) was also negatively associated with cognitive development (-0.31 (95% CI: -0.49, -0.13)), while the HOME inventory, an index of quality of the home environment (0.46 (95% CI: 0.21, 0.72)) and socio-economic status (1.50 (95% CI: 1.03, 1.98)) were positively associated with cognitive development.

**Data Availability Statement:** The data are not publicly available due to ethical restrictions on participant privacy. Data for this study are available upon request to others in the scientific community.

For access, please contact David Spiro (david.
spiro@nih.gov).

**Funding:** The Etiology, Risk Factors and
Interactions of Enteric Infections and Malnutrition
and the Consequences for Child Health and
Development Project (MAL-ED) is a collaborative
project supported by the Bill & Melinda Gates
Foundation, the Foundation for the National
Institutes of Health, and the National Institutes of
Health, Fogarty International Center. The funders
had no role in study design, data collection and
analysis, decision to publish, or preparation of the
manuscript.

**Competing interests:** The authors have declared
that no competing interests exist.

**Abbreviations:** AGP, α-1-acid glycoprotein; ALRI,
Acute lower respiratory infection; BGD, Dhaka,
Bangladesh; BRF, Fortaleza, Brazil; BRINDA,
Biomarkers Reflecting Inflammation and Nutrition
Determinants of Anemia; CI, Confidence interval;
EBF, Exclusive breastfeeding; FR, Fluid reasoning;
HAZ, Height for age z score; HOME, Home
Observation for Measurement of the Environment;
HR, Hazard ratio; INV, Vellore, India; LMICs, Low
and middle-income countries; MAL-ED, The
Etiology, Risk Factors and Interactions of Enteric
Infections and Malnutrition and the Consequences
for Child Health and Development; NEB, Bhaktapur,
Nepal; SAV, Venda, South Africa; SD, Standard
Deviation; TfR, Transferrin receptor; TZH, Haydom,
Tanzania; WAMI, Water and sanitation, assets,
maternal education, and income; WPPSI, The
Wechsler Preschool Primary Scales of Intelligence.

## Conclusions

Early-onset persistent stunting was associated with lower cognitive development in children at 5 years of age in this cohort of children.

## Introduction

Cognitive development in early childhood involves the development of thinking, attention, memory, and problem solving, all of which help children to understand the world around them [1]. The foundation of adult health and wellbeing is grounded on early childhood development, which is included as a necessary component of Sustainable Development Goal-4 [2]. Proxy measures of stunting and poverty indicate that in low and middle-income countries (LMICs) an estimated 250 million children under five (43%) are at risk of not reaching their developmental potential [3], which can lead to loss of adult income as well as poor national development [4]. The social and economic growth of a country can benefit from ensuring the optimal cognitive development of children, and this may also help to break the intergenerational poverty cycle in LMICs [5].

In 2018, an estimated 151 million children under five (22%) were stunted globally [6], particularly in LMICs [7]. Stunting is believed to contribute to children's poor cognitive development, behavioral problems and poor school achievement that can persist through adulthood [4, 8–10]. Linear growth failure in childhood is the most prevalent form of undernutrition globally [11]. Stunting at two years of age has been shown to be associated with cognitive and psychosocial outcomes in late childhood [10, 12]. A study conducted in India reported that stunted children had significantly poorer performance on short-term memory, retrieval ability and visual-spatial ability tests [13]. Other studies in the Philippines and South Africa reported that children who developed stunting and never exhibited catch up growth (i.e., persistently stunted children) had poorer cognitive scores [12, 14].

The association between persistent stunting and cognitive development has been demonstrated [15]. A recent meta-analysis of 29 LMICs has established a positive association between linear growth and cognitive development in the first 2 years of life [16], but the meta-analysis could not integrate environmental, educational, and follow-up data to describe the relationship between stunting and cognitive impairment. Previous longitudinal studies have considered the relationship between linear growth and cognitive development [17–24]; however, the MAL-ED study collected an unprecedented range of data on the demographics and exposures of children over time and will be able to explore this question while controlling for potential confounding factors. We investigated whether the timing and persistence of stunting plays a role in long-term cognitive development using data from the MAL-ED study. Our aim was to evaluate the association between early-onset versus later-onset persistent stunting and cognitive development at 60 months of age, and subsequently to identify the factors associated with early-onset stunting.

## Methods

### Study setting and participants

The Etiology, Risk Factors and Interactions of Enteric Infections and Malnutrition and the Consequences for Child Health and Development (MAL-ED) study is a prospective birth cohort study which was implemented in eight different study sites [25]. Data were collected

between November 2009 and February 2017 in Dhaka, Bangladesh (BGD); Fortaleza, Brazil (BRF); Vellore, India (INV); Bhaktapur, Nepal (NEB); Loreto, Peru (PEL); Naushero Feroze, Pakistan (PKN); Venda, South Africa (SAV), and Haydom, Tanzania (TZH). The sample size for MAL-ED (at least 200 children per site followed for two years) was calculated for the primary aims of the study, which involved the relationship between enteric infection and growth and development and the results have been published elsewhere [26–28]. The sites had different enrollment procedures; some sites enrolled all available and eligible children in order to meet the 10–12 per month site recommendation, whereas others randomly selected children from the pool of eligible children in their population [29–36]. Children were followed up to five years of age as an extension of the study. Eligible infants were less than 17 days old, born singleton with a birth weight > 1500 g, without serious illnesses, to a mother at least 16 years of age, and to a family intending to stay in the community for at least 6 months. In the original study, 1565 infants were enrolled from the six sites included in these analyses; the two sites excluded here were Loreto, Peru (cognitive data were not available from the Peru site at 5 years) and Naushero Feroze, Pakistan (due to bias in the measurements of length in a subset of participants in the Pakistan site).

Each site obtained ethical approval from the appropriate authorities of their respective institutions, and written informed consent was obtained from the mothers or caregivers of the participants. Additional ethical approvals were obtained for the extension study in each of the sites and a second signed consent was obtained from the participants where necessary.

## Outcome variable

The primary outcome for this research is the Wechsler Preschool Primary Scale of Intelligence (WPPSI) III score [37] which was used to assess cognitive abilities at 5 years. The details of the WPPSI score have already been published [38]. Several cultural modifications were needed in the subtests used for multisite analyses, which include (a) modification of WPPSI items including pictures to fit local cultural norms, (b) translation and back-translation by experts in both English and the local language, (c) changes in which subtests were administered following field testing and (d) changes following pilot testing [39]. The scales were adapted to account for cultural appropriateness, administered by trained assessors, and around 8% were recorded and reviewed for quality control purposes. These scores were not standardized for comparison with the US reference population since our sample characteristics are distinct. A single factor with support across all sites describing fluid reasoning was identified through psychometric analyses [39]. Fluid reasoning (FR) is the capacity to think logically and solve problems in novel situations, independent of acquired knowledge [40]. Three subscales (block design, matrix reasoning, and picture completion) were used to develop this factor. This was then converted to T-scores, which means that data from all sites were combined and standardized to a mean of 50 and SD of 10 [37]. Cognitive development scores have not been reported here due to historical abuse [41].

## Key explanatory variable

Stunting is defined as length/height-for-age z-score (HAZ) below minus two standard deviations (SD) from the median of the reference population, using the WHO Child Growth Standards [42]. Stunting was categorized as early-onset persistent (first stunted at 1–6 months and persisting at 60 months), early-onset only (first stunted at 1–6 months and not stunted at 60 months), late-onset persistent (first stunted at 7–24 months and persisting at 60 months), late-onset only (first stunted at 7–24 months and not stunted at 60 months), and never (never stunted).

## Socioeconomic status and environment inventory

Mothers or caregivers were questioned every 6 months about their sources of water and sanitation facilities, assets, and income. Information on maternal education was collected at enrollment. We combined water and sanitation, assets, maternal education, and income into a composite score (WAMI index) [43]. As the WAMI index showed little variability over time, we averaged their values across all collected time points.

The Home Observation for Measurement of the Environment (HOME) Inventory is a widely recognized and commonly used instrument to evaluate the quality of the home environment. The HOME has been frequently revised and adapted for a variety of contexts. Versions of the HOME intended for younger children have been validated within this sample [44]. The HOME Inventory was used at 60 months to capture dimensions of the home environment beneficial for child development. One factor supported by psychometric analysis survey at age 60 months was examined, describing the behaviors and environment to support learning around the child. These behaviors include encouragement to learn numbers, shapes, the alphabet, and colors, and to learn to read. In addition, this survey evaluates the environment, in that the child has resources (puzzles, audio players, toys), that the living space is safe and not too dark or crowded, and that the parent responds verbally and supportively to the child.

## Morbidity

During the first two years of life, daily information on child illnesses and antibiotic use was collected from the mother at twice weekly home visits. We evaluated 3 common illnesses—diarrhea, acute lower respiratory infection (ALRI), and fever and, as detailed elsewhere, using standard definitions of illness onset and episodes [45]. We calculated the longitudinal incidence of each of these as the cumulative number of episodes divided by the total number of follow-up days and multiplied by 365 days (e.g., illness episodes per year).

## Breastfeeding practices

Initiation of breastfeeding was recorded at enrollment and mothers were queried about the provision of colostrum and the use of pre-lacteal feedings. As part of the twice weekly surveillance visits, caregivers were queried about breastfeeding and use of non-breast milk liquids and solids. In between visits, we assumed that breastfeeding status did not change in order to calculate the duration of each breastfeeding practice in days [46]. We defined breastfeeding status as exclusive if the child received only breast milk with the exception of vitamins or medicine.

## Micronutrient status

Blood samples were collected to evaluate iron, zinc and vitamin A status and anemia at 7, 15 and 24 months [47]. The concentration of plasma α-1-acid glycoprotein (AGP) was used to adjust the biomarker concentrations using the Biomarkers Reflecting Inflammation and Nutrition Determinants of Anemia (BRINDA) method [48]. For analyses, we averaged each of the biochemical indicators from all available blood draws for each child and used square-root transformations to normalize their distributions where appropriate. Plasma transferrin receptor (TfR) has been found to be important for both cognitive development and growth in the MAL-ED cohort; therefore, we included it in this analysis [37, 49].

## Microbiology

Non-diarrheal stool samples were collected monthly (during the first year of life) and quarterly (during the second year of life). The samples were tested for more than 40 enteropathogens

[50]. We calculated the per-child bacterial and viral load, defined as the average number of bacteria and viruses identified in non-diarrheal stool samples during first two years of life.

## Biomarkers of Environmental Enteric Dysfunction (EED)

Fecal concentrations of myeloperoxidase and alpha-1-antitrypsin were measured on non-diarrheal stool samples as markers of gut permeability and inflammation. Each of these fecal biomarkers was log-transformed, and then a regression was performed in order to de-trend for age, recently reported breast milk consumption or fever, time between collection and sample testing, and stool consistency [51].

## Statistical analyses

Linear mixed models were used to estimate the relationship between stunting status and the cognitive development at 60 months, including random intercept for study site. To adjust for clustering by site, we included site as a random effect. This approach allows robust estimation of variance in the outcome variable within and between the clusters [52]. Stunting is associated with a number of other factors which are also correlated with cognitive development. We included these factors in the model to minimize confounding effects. We estimated four models for our outcome variable, introducing additional controls at each stage (Model 1: unadjusted (site); Model 2: adjusted for child's sex and birth weight; Model 3: plus exclusive breast feeding duration, morbidity, gut inflammation and enteropathogen; and Model 4: plus socioeconomic status). Also we have tested for interaction and there was no evidence of interaction in terms of site and other variables.

To identify the factors associated with early-onset stunting, survival analysis with mixedeffect exponential hazards regression model was carried out with site as a random effect. Bivariate associations between each independent variable with risk of early-onset stunting were determined using unadjusted mixed-effect exponential hazards regression models, and variables associated with risk of early-onset stunting at the level of $p \leq 0.2$ were included in the multivariable model. The strength of association was determined by calculating hazard ratio (HR) and their 95% confidence intervals (CIs). All of the mixed-effects data analyses were conducted using the lme4 package in R version 3.5.1 (R Foundation). The mixed-effect exponential hazards regression models were run in Stata version 15.1 (College Station, TX, USA).

## Ethics

Ethical approval was obtained from the institutional review boards at participating institutions (S1 Table). Written informed consent was obtained from the parent or guardian of each participating child.

## Results

We enrolled 1,565 children across six research sites in the MAL-ED study (Bangladesh, India, Nepal, Brazil, South Africa and Tanzania), of whom 968 children had WPPSI data at 60±2 months and height data at 57–60 months, and 943 children had complete data considering all other variables (Fig 1).

Among the 288 children who were stunted in the first six months of life, 48% were no longer stunted at 5 years of age. For the 256 children with late-onset stunting, 63% were no longer stunted at 5 years of age. Fifty percent of children were female, average birth weights ranged from 2.82 kg in BGD to 3.33 kg in BRF, and median exclusive breastfeeding (EBF) duration ranged from 25 days in SAV to 107 days in BGD (Table 1).

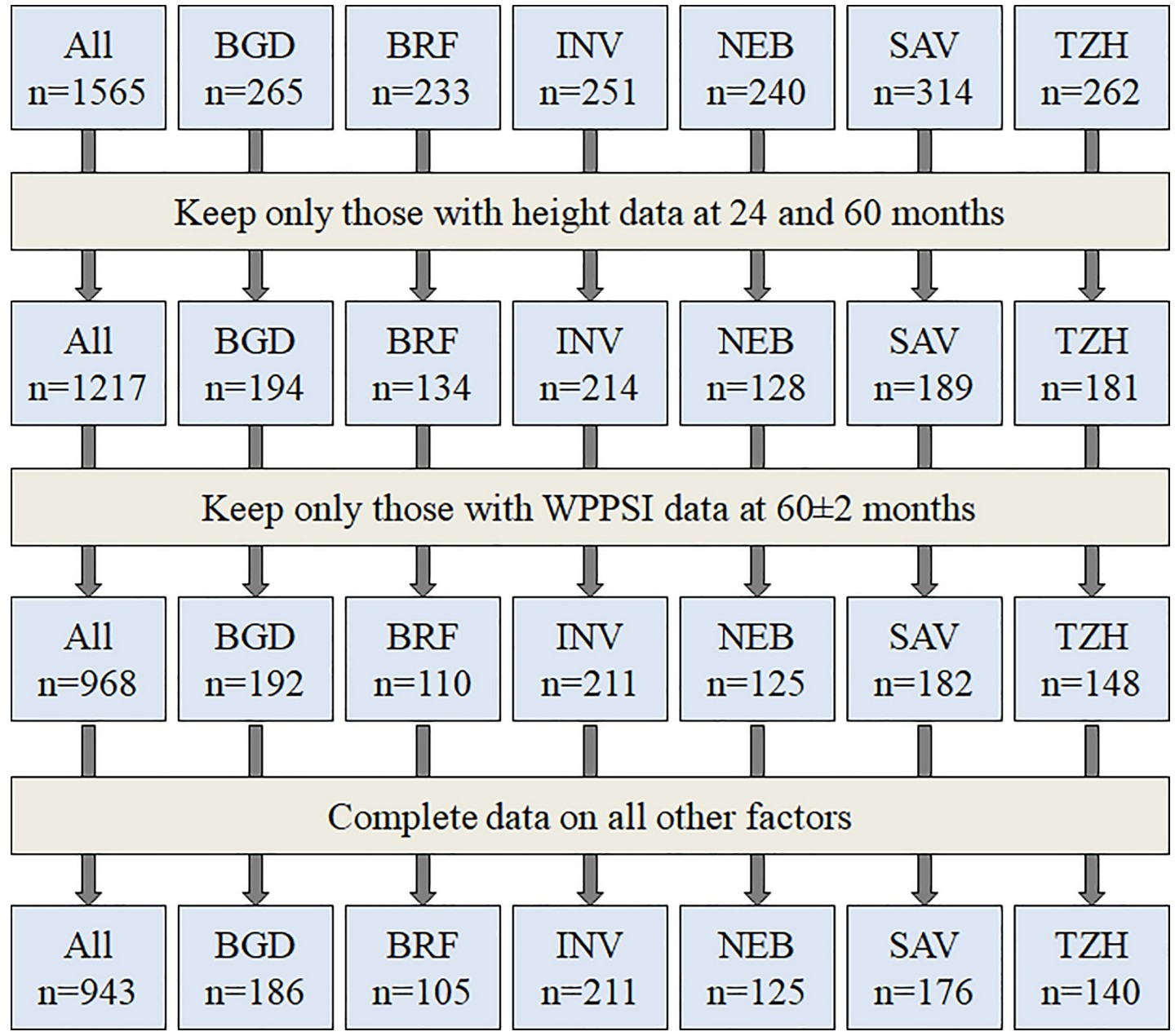

**Fig 1. MAL-ED cohort profile of children who were retained in the analysis.** [Sites: BGD: Bangladesh–Dhaka; BRF: Brazil–Fortaleza; INV: India–Vellore; NEB: Nepal–Bhaktapur; SAV: South Africa–Venda; TZH: Tanzania–Haydom].

### Stunting and cognitive development

The unadjusted estimates for Model 1 suggest that early-onset persistently stunted children had statistically significantly lower cognitive scores than those who were never stunted, and this remained true when adjusting for confounding factors in Model 4 (Table 2). Model 1 also shows that the early-onset only group had a lower cognitive status than the comparison group (never stunted), but this difference was no longer statistically significant when the model was adjusted for other factors. No significant association was found between late-onset stunting and WPPSI scores at 5 years.

**Table 1. Characteristics of the study sample.**

| Indicators | Overall (n = 943) | BGD (n = 186) | BRF (n = 105) | INV (n = 211) | NEB (n = 125) | SAV (n = 176) | TZH (n = 140) |
|---|---|---|---|---|---|---|---|
| Stunting status, n (%) | | | | | | | |
| Never | 399 (42) | 71 (38) | 85 (81) | 79 (37) | 81 (65) | 65 (37) | 18 (13) |
| Early-onset persistent | 150 (16) | 35 (19) | 0 (0) | 42 (20) | 14 (11) | 21 (12) | 38 (27) |
| Early-onset only | 138 (15) | 23 (12) | 16 (15) | 32 (15) | 7 (6) | 49 (28) | 11 (8) |
| Late-onset persistent | 95 (10) | 24 (13) | 2 (2) | 19 (9) | 16 (13) | 5 (3) | 29 (21) |
| Late-onset only | 161 (17) | 33 (18) | 2 (2) | 39 (18) | 7 (6) | 36 (20) | 44 (31) |
| Male children, n (%) | 469 (50) | 95 (51) | 56 (53) | 98 (46) | 57 (46) | 89 (51) | 74 (53) |
| Birth weight (kg), mean ± SD | 3.05 ± 0.48 | 2.82 ± 0.41 | 3.33 ± 0.50 | 2.89 ± 0.45 | 3.00 ± 0.41 | 3.15 ± 0.46 | 3.30 ± 0.44 |
| Exclusive breastfeeding days, median (IQR) | 53 (24, 97) | 107 (58, 154) | 60 (28, 110) | 78 (39, 112) | 36 (18, 89) | 25 (13, 40) | 38 (20, 68) |
| Incidence of ALRI (0-24m), mean ± SD | 0.67 ± 1.02 | 0.48 ± 0.76 | 0.10 ± 0.24 | 1.56 ± 1.38 | 0.85 ± 1.12 | 0.30 ± 0.51 | 0.30 ± 0.37 |
| Incidence of diarrhea (0-24m), mean ± SD | 1.87 ± 1.93 | 3.65 ± 2.34 | 0.54 ± 0.58 | 2.09 ± 1.73 | 2.37 ± 1.81 | 0.64 ± 0.83 | 1.23 ± 1.13 |
| Adjusted transferrin receptor (mcg/mL), mean ± SD | 6.58 ± 3.55 | 7.46 ± 3.38 | 9.81 ± 2.51 | 5.34 ± 3.07 | 9.51 ± 3.85 | 4.53 ± 2.07 | 4.82 ± 2.31 |
| α-1 antitrypsin (mg/g) in log10 scale, mean ± SD | -1.20 ± 0.45 | -0.95 ± 0.37 | -1.25 ± 0.48 | -1.11 ± 0.38 | -0.93 ± 0.33 | -1.53 ± 0.35 | -1.46 ± 0.46 |
| Myeloperoxidase (ng/mL) in log10 scale, mean ± SD | 8.42 ± 0.56 | 8.31 ± 0.38 | 7.80 ± 0.75 | 8.86 ± 0.47 | 8.23 ± 0.44 | 8.39 ± 0.39 | 8.58 ± 0.41 |
| Bacteria load, mean ± SD | 0.80 ± 0.32 | 0.90 ± 0.25 | 1.04 ± 0.32 | 0.79 ± 0.28 | 0.64 ± 0.22 | 0.50 ± 0.20 | 1.04 ± 0.31 |
| Virus load, mean ± SD | 0.06 ± 0.07 | 0.08 ± 0.08 | 0.03 ± 0.07 | 0.06 ± 0.06 | 0.05 ± 0.05 | 0.05 ± 0.06 | 0.09 ± 0.08 |
| HOME behaviors and environment score, mean ± SD | 11.22 ± 3.20 | 10.37 ± 1.9 | 9.61 ± 2.9 | 14.69 ± 1.88 | 12.63 ± 2.12 | 10.98 ± 1.97 | 7.58 ± 2.73 |
| WAMI index, mean ± SD | 0.59 ± 0.22 | 0.57 ± 0.12 | 0.83 ± 0.08 | 0.50 ± 0.14 | 0.71 ± 0.13 | 0.76 ± 0.10 | 0.23 ± 0.10 |

Sites: BGD: Bangladesh—Dhaka; INV: India—Vellore; NEB: Nepal—Bhaktapur; BRF: Brazil—Fortaleza; SAV: South Africa—Venda; TZH: Tanzania—Haydom; SD: Standard deviation; IQR: Interquartile range; ALRI: Acute lower respiratory infection; WPPSI: The Wechsler Preschool Primary Scales of Intelligence; HOME: Home Observation for Measurement of the Environment; WAMI: Water and sanitation, assets, maternal education, and income score

## Other factors and cognitive development

Transferrin receptor and bacterial load showed statistically significant negative associations with cognitive development (Model 3). Bacterial load was no longer associated with WPPSI score when HOME inventory and WAMI index were included in the model, whereas TfR remained statistically significant in Model 4. HOME inventory and WAMI index were positively associated with cognitive development in Model 4. No associations were found with child's sex, birth weight, EBF, ALRI, diarrheal infection, α-1 antitrypsin, myeloperoxidase or viral load (Table 2).

## Factors associated with early stunting

Because early-onset persistent stunting was found to be negatively associated with WPPSI score, we conducted mixed-effect exponential hazards regression analysis to determine the factors associated with early-onset stunting. The mixed-effect exponential hazards regression model showed that female sex, birth weight, treatment of drinking water, family income, and maternal height were significantly associated with risk of stunting, but the provision of colostrum, maternal education and age, and type of sanitation were not (Table 3). The South African and Tanzanian sites had the earliest development of stunting (Fig 2).

## Discussion

In this study we have shown that early-onset persistent stunting in children is negatively associated with cognitive development at five years of age when compared to those who were never stunted. This relationship remained significant after adjusting for potential confounding

**Table 2. Multivariate mixed effect linear regression model results of factors associated with WPPSI t-scores at five years of age at six sites in the MAL-ED study.**

|  | Model 1 | Model 2 | Model 3 | Model 4 |
|---|---|---|---|---|
| **Key explanatory variable** |  |  |  |  |
| Stunting status (Reference: Never) |  |  |  |  |
| Early-onset persistent | -3.53 (-5.16, -1.90)*** | -2.93 (-4.67, -1.18)** | -3.03 (-4.82, -1.24)** | -2.10 (-3.85, -0.35)* |
| Early-onset only | -1.66 (-3.31, -0.01)* | -1.06 (-2.82, 0.70) | -0.98 (-2.76, 0.80) | -1.21 (-2.93, 0.52) |
| Late-onset persistent | -1.12 (-3.07, 0.82) | -1.08 (-3.03, 0.86) | -1.37 (-3.34, 0.60) | -0.68 (-2.60, 1.24) |
| Late-onset only | -1.40 (-3.01, 0.21) | -1.27 (-2.89, 0.35) | -1.24 (-2.90, 0.42) | -1.09 (-2.70, 0.53) |
| **Birth** |  |  |  |  |
| Child sex, male |  | -0.35 (-1.45, 0.75) | 0.05 (-1.08, 1.17) | 0.09 (-1.01, 1.18) |
| Birth weight in kg |  | 1.30 (-0.02, 2.62) | 0.84 (-0.51, 2.19) | 0.70 (-0.61, 2.00) |
| **Breast feeding** |  |  |  |  |
| EBF duration (months) |  |  | -0.13 (-0.50, 0.25) | -0.02 (-0.38, 0.34) |
| **Morbidity** |  |  |  |  |
| Incidence of ALRI (0–24 months) |  |  | -0.05 (-0.67, 0.57) | 0.34 (-0.27, 0.94) |
| Incidence of diarrhea (0–24 months) |  |  | 0.01 (-0.34, 0.35) | 0.04 (-0.29, 0.37) |
| **Systemic inflammation** |  |  |  |  |
| Transferrin receptor (mcg/mL) |  |  | -0.35 (-0.53, -0.16)*** | -0.31 (-0.49, -0.13)* |
| **Gut inflammation** |  |  |  |  |
| α-1 antitrypsin (mg/g) in log10 scale |  |  | 1.00 (-0.50, 2.49) | 1.02 (-0.43, 2.46) |
| Myeloperoxidase (ng/mL) in log10 scale |  |  | -0.60 (-1.84, 0.64) | -0.27 (-1.47, 0.93) |
| **Enteropathogen load** |  |  |  |  |
| Bacteria load (10% increase) |  |  | -0.22 (-0.43, -0.01)* | -0.06 (-0.26, 0.15) |
| Virus load (10% increase) |  |  | -0.27 (-1.05, 0.50) | -0.28 (-1.03, 0.47) |
| **Socio-economic and HOME Inventory** |  |  |  |  |
| HOME behaviors and environment score |  |  |  | 0.46 (0.21, 0.72)* |
| WAMI index (10% increase) |  |  |  | 1.50 (1.03, 1.98)* |
| Conditional R$^2$ | 0.36 | 0.37 | 0.41 | 0.39 |

***p < 0.001

**p < 0.01

*p < 0.05

EBF: Exclusive breastfeeding; ALRI: Acute lower respiratory infection; HOME: Home Observation for Measurement of the Environment; WAMI: Water and sanitation, assets, maternal education, and income score

**Table 3. Estimates of hazard ratios (HR) and 95% confidence intervals (CI) of first time stunting status using mixed-effects exponential proportional hazards regression model.**

| Characteristic | Unadjusted HR (95% CI) | p-value | Adjusted HR (95% CI) | p-value |
|---|---|---|---|---|
| Child sex, male | 1.55 (1.29, 1.87) | <0.001 | 1.78 (1.47, 2.16) | <0.001 |
| Birth weight in kg | 0.46 (0.36, 0.58) | <0.001 | 0.43 (0.33, 0.55) | <0.001 |
| Colostrum given | 0.69 (0.46, 1.05) | 0.081 | 0.83 (0.55, 1.25) | 0.370 |
| Prelacteal feeding | 0.84 (0.57, 1.23) | 0.383 |  |  |
| Treatment of drinking water | 0.56 (0.42, 0.73) | <0.001 | 0.63 (0.48, 0.84) | 0.002 |
| Improved drinking water | 1.17 (0.83, 1.65) | 0.371 |  |  |
| Improved sanitation | 0.70 (0.51, 0.98) | 0.038 | 0.72 (0.51, 1.00) | 0.053 |
| Family income ($100 increase) | 0.90 (0.83, 0.97) | 0.007 | 0.93 (0.86, 0.99) | 0.049 |
| Maternal education in years | 0.97 (0.94, 0.99) | 0.046 | 0.99 (0.96, 1.02) | 0.673 |
| Maternal age in years | 0.98 (0.97, 1.00) | 0.058 | 0.99 (0.97, 1.00) | 0.160 |
| Maternal height in cm | 0.95 (0.94, 0.97) | <0.001 | 0.96 (0.94, 0.97) | <0.001 |

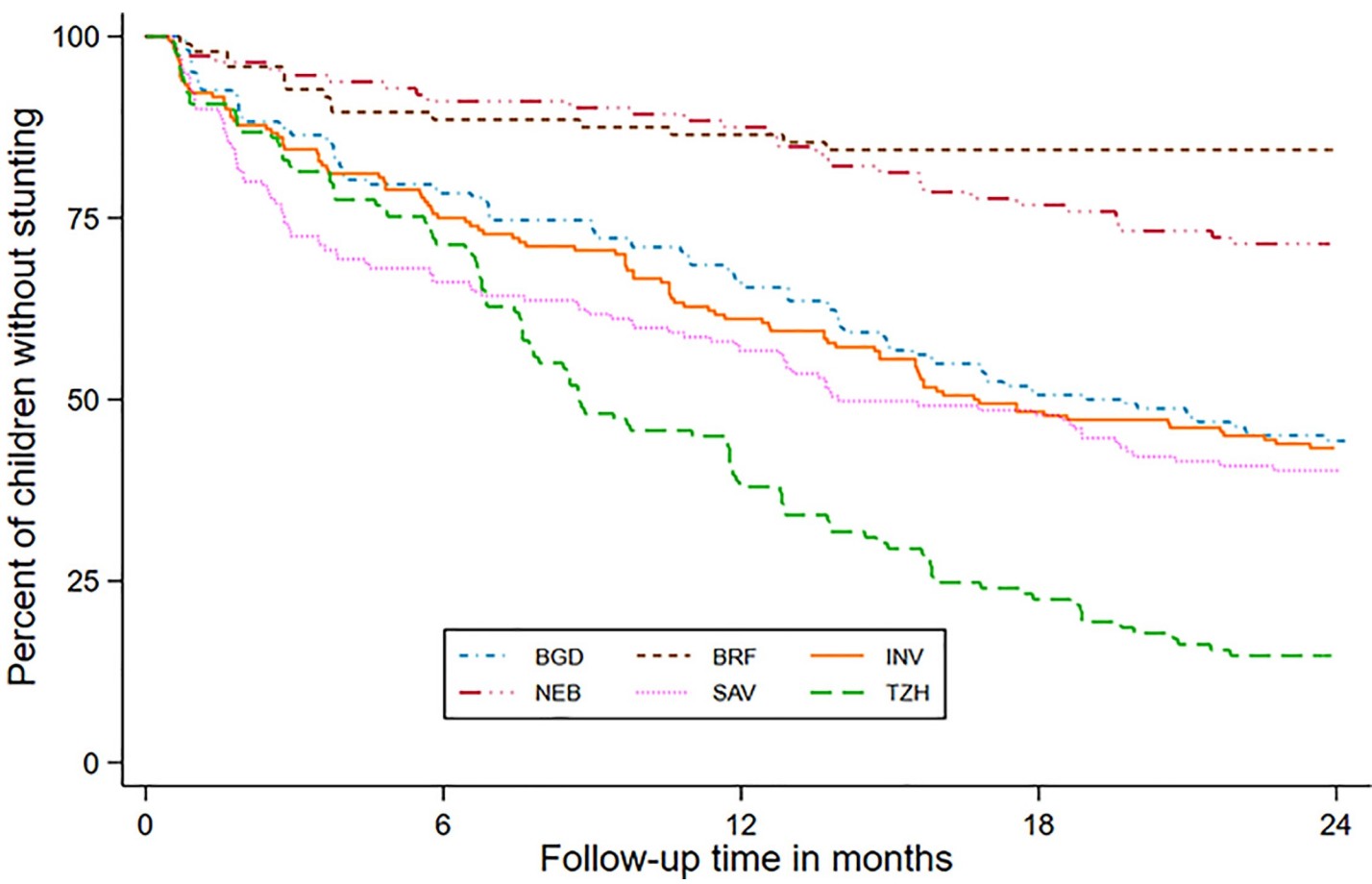

**Fig 2. Percent of children who had not yet experienced stunting through 2 years of age at six sites (BGD: Bangladesh—Dhaka; INV: India—Vellore; NEB: Nepal—Bhaktapur; BRF: Brazil—Fortaleza; SAV: South Africa—Venda; TZH: Tanzania—Haydom).**

factors. Although the relationship was statistically significant only for the early-onset persistent group, the point estimates for the other stunted groups were all negative compared with those who were never stunted.

The relationship between stunting during early childhood and cognitive function in late childhood is well established. Exposure to biological and psychosocial risk factors such as stunting, poverty, and poor home environment have been associated with poor cognitive development in early childhood [3].

The results of the present study are similar to other studies in the literature that have considered timing and length of stunting in relation to cognitive development. In one Peruvian study, children who were persistently stunted from age 6 months to 6 years or stunted in childhood (late onset, during 4.5–6 years) had significantly lower scores on cognitive skills (verbal vocabulary and quantitative test scores) at 4.5 to 6 years, compared to the non-stunted peers. However, the children who were no longer stunted did not differ from the never stunted children on both assessments [53]. In another Peruvian study, Berkman *et al.* documented lower cognitive scores at 9 years of age in children who were persistently stunted from birth to 2 years of age compared to those who were never stunted during this period. Recovered and late incident children did not significantly differ from those who were never stunted [54]. In a Filipino study, Mendez and Adair found significantly lower cognition scores at 8 and 11 years in persistently stunted children than in non-stunted children. In this study, children who were

stunted and recovered also had significantly lower scores than those who were never stunted [12]. Although these studies considered different periods of exposure, they consistently found that persistently stunted children scored lower on assessments than did children who were never stunted; however, these children likely also suffer a number of deprivations that contribute to their cognitive development.

We observed a negative relationship between early life bacterial and viral pathogen burden with cognitive development at 5 years but it was not statistically significant. However, previously it was shown that cognitive development at 24 months of age was negatively affected by enteropathogen detection rates [26]. Similar findings were observed for the data on illnesses. The temporality of our assessments likely plays a factor in the observed significance of these relationships at 24 months but not at 5 years, in that risk factors may change in the intervening three years. In addition, the study may not have been powered to observe subtle associations or infrequent risk factors. Interestingly, our study-confirmed symptoms of illness were not related to cognitive development at 5 years, only maternally reported symptoms of illness were. It is possible that mothers change their behavior as a result of perceived child illness and this may influence the relationship between illness and cognitive development [37].

We found that higher concentration of plasma TfR was negatively associated with the cognitive development at 5 years. Although not focused on TfR concentrations, previous studies have indicated a relationship between iron deficiency anemia during early childhood and poor development of cognitive functions in later life [55, 56]. TfR, a known marker of iron deficiency, is usually affected by inflammation, and therefore, we have adjusted it with a marker of inflammation. Evidence suggests that TfR is associated with cellular proliferation, erythropoiesis and rapid growth of children, particularly in the early years of life [57]. TfR can be expressed in red blood cells and the brain, and according to previous literature it prioritizes red blood cells over the brain [58]. Although the mechanism is not yet elucidated, it is possible that due to increased expression of TfR on red blood cell precursors there may be a simultaneous decrease in TfR expression in the brain due to prioritization of red cells over brain [58]. We have measured the TfR in peripheral blood, hence the result is indicative of increased expression of TfR on red cell precursors among the children of this study.

Our analyses showed that WAMI and WPPSI scores were positively associated at 5 years. The association between socioeconomic status and child development is well established [4, 59, 60]. Socioeconomic factors affect brain development through different mechanisms, such as prenatal factors, parental care, cognitive stimulation, toxin exposure, nutrition and stress [61, 62]. Lower socioeconomic status has also been found to be associated with poor nutritional status, sanitary and hygiene conditions, which in turn have been associated with higher infection rates and stunting in children. All these factors are known to contribute to childhood development [63, 64]. In addition, we found that HOME inventory was positively associated with the cognitive development at 5 years.

After determining that early-onset persistent stunting was negatively associated with cognitive development at 60 months, we identified several factors that were protective against the risk of stunting, for example being female, higher birth weight, treatment of drinking water, higher family income, and taller maternal height. These protective factors, while not entirely new [28, 65–77], could help identify specific groups to target with an intervention because they might be at greater risk, primarily lower socioeconomic mothers, ideally with prenatal interventions that would improve both socioeconomic status and prenatal nutrition. The associations with birth weight and maternal height indicate that programs solely focused on children may have limited impact as pre-natal and intergenerational forces are influential.

This study has several limitations. First, the study was conducted in 6 LMICs among children who hailed from limited resource settings. Comparison with children from high income

settings would enable a better understanding of the full negative effect of persistent stunting on cognitive development. Moreover, we did not have data on certain variables which are important predictors of poor cognitive functions in children during their early years of life, for example, maternal health, lack of child stimulation, exposure to violence and some environmental factors. In addition, some of the factors included in the model may be associated with stunting status and including them as covariates may lead to over-adjustment. More complex models may be planned in the future to account for interrelationships among the different risk factors. Finally, many of the children were either lost to follow up or the extension funding arrived too late for the children to participate in the five year follow up. Although we compared those who were lost to follow up with those were retained and did not find many differences, it is possible that the children who were retained in the study were different in some ways (S2 Table). Despite these limitations, the study was well designed and equipped with skilled staff and high-quality laboratory facilities. To our knowledge this is the first attempt to address the role of persistent stunting on the cognitive development of children living in 6 different countries across 3 continents. Moreover, the inclusion of multi-country data has enhanced the quality of analysis as well as strength of the findings.

In summary, early-onset persistent stunting and higher TfR concentrations were associated with lower cognitive development scores at 5 years of age in this cohort of children. Socio-economic status was also found to be positively associated with cognitive development at 5 years of age. It implies the importance of potential programs to address the issue of persistent stunting and iron deficiency in the early years of life. Initiatives to be taken to combat early stunting include improving birth weight, socio-economic status, and safe water treatment practices in the household.

## Supporting information

**S1 Checklist. STROBE Statement—Checklist of items that should be included in reports of cohort studies.**
(DOCX)

**S1 Table. Institutional Research Boards Approvals.**
(DOCX)

**S2 Table. Comparison of children retained in the cohort through five years of age ('Include') and those lost-to-follow up or with incomplete data ('Exclude').**
(DOCX)

## Acknowledgments

We are grateful to the children and caregivers who participated in the study for their invaluable contributions.

## Author Contributions

**Conceptualization:** Md Ashraful Alam, Stephanie A. Richard, Shah Mohammad Fahim, Mustafa Mahfuz, Subhasish Das, Binod Shrestha, Beena Koshy, Estomih Mduma, Jessica C. Seidman, Laura E. Murray-Kolb, Laura E. Caulfield, Tahmeed Ahmed.

**Data curation:** Md Ashraful Alam, Binod Shrestha, Beena Koshy, Estomih Mduma.

**Formal analysis:** Md Ashraful Alam, Stephanie A. Richard, Shah Mohammad Fahim, Baitun Nahar, Jessica C. Seidman.

**Funding acquisition:** Md Ashraful Alam.

**Investigation:** Md Ashraful Alam, Stephanie A. Richard, Mustafa Mahfuz, Baitun Nahar, Subhasish Das, Binod Shrestha, Beena Koshy, Estomih Mduma, Laura E. Murray-Kolb, Laura E. Caulfield, Tahmeed Ahmed.

**Methodology:** Md Ashraful Alam, Stephanie A. Richard, Mustafa Mahfuz, Subhasish Das, Binod Shrestha, Beena Koshy, Estomih Mduma, Laura E. Murray-Kolb, Laura E. Caulfield.

**Project administration:** Baitun Nahar, Laura E. Murray-Kolb, Tahmeed Ahmed.

**Software:** Md Ashraful Alam, Jessica C. Seidman.

**Supervision:** Stephanie A. Richard.

**Visualization:** Md Ashraful Alam, Stephanie A. Richard.

**Writing – original draft:** Md Ashraful Alam.

**Writing – review & editing:** Md Ashraful Alam, Stephanie A. Richard, Shah Mohammad Fahim, Mustafa Mahfuz, Baitun Nahar, Subhasish Das, Binod Shrestha, Beena Koshy, Estomih Mduma, Jessica C. Seidman, Laura E. Murray-Kolb, Laura E. Caulfield, Tahmeed Ahmed.

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
