## [Decision Letter · Decision Letter 0]

13 Nov 2019

PONE-D-19-28331

Impact of early-onset persistent stunting on cognitive development at 5 years of age: Results from a multi-country cohort study

PLOS ONE

Dear Dr Mahfuz,

Thank you for submitting your manuscript to PLOS ONE. After careful consideration, we feel that it has merit but does not fully meet PLOS ONE’s publication criteria as it currently stands. Therefore, we invite you to submit a revised version of the manuscript that addresses the points raised during the review process.

The paper presents the finding of a rigorous research on the relationship between early onset stunting and cognitive development of children from multicentre cohort study. The research question is well formulated and supported with literature, detailed and extensive data had been collected and presented on many essential confounders, and statistical adjustment was made using a robust model. The manuscript is also well-written and the methods and findings are clearly presented. However, as pointed out below, the reviewers have raised important concerns on the possibility of over adjustment bias and the need of providing additional clarification on how some of the variables were measured.  

We would appreciate receiving your revised manuscript by Dec 28 2019 11:59PM. To enhance the reproducibility of your results, we recommend that if applicable you deposit your laboratory protocols in protocols.io, where a protocol can be assigned its own identifier (DOI) such that it can be cited independently in the future. For instructions see: http://journals.plos.org/plosone/s/submission-guidelines#loc-laboratory-protocols

We look forward to receiving your revised manuscript.

Kind regards,

Samson Gebremedhin, PhD

Academic Editor

PLOS ONE

"Ethics approval was obtained from the institutional review boards at participating institutions (Supplement Table 1). Written informed consent was obtained from the parent or guardian of each participating child.".

i) Please amend your current ethics statement to include the full name of the ethics committee/institutional review board(s) that approved your specific study.

ii) Once you have amended this/these statement(s) in the Methods section of the manuscript, please add the same text to the “Ethics Statement” field of the submission form (via “Edit Submission”).

"This study was supported by the Bill and Melinda Gates Foundation. The funders had

no role in study design, data collection and analysis, decision to publish, or preparation

of the manuscript.".

i) Please provide an amended statement that declares *all* the funding or sources of support (whether external or internal to your organization) received during this study, as detailed online in our guide for authors at http://journals.plos.org/plosone/s/submit-now.  Please also include the statement “There was no additional external funding received for this study.” in your updated Funding Statement.

ii) Please include your amended Funding Statement within your cover letter. We will change the online submission form on your behalf.

Additional Editor Comments (if provided):

Abstract: Please state in bracket that the HOME inventory is an index of quality of the home environment. In the existing form it may confuse readers.

Background 87-88: “However, the meta-analysis was based on observational studies……”????? I don’t think it would be possible to determine the relationship between linear growth and cognitive development through experimental design. It is also important that the current study also followed observational design.

Methods:

106-7: was there any statistical basis for planning 200 children per site? Also, I see nothing about the sampling approach used for enrolling the children at baseline.

Line 145-151: please add a sentence or two that describes the sub-scales of the “HOME inventory”.

I fear AGP measurement alone (I,e, without CRP) may not fully adjust the micronutrient markers for inflammation. It is not clear why the authors did not collect data of CRP too.

Discussion

Can you please discuss the possibility of over-adjustment bias in the model because impaired immunity and increased susceptibility to infection can be a pathway that links stunting with psychomotor development.

Reviewers' comments:

Reviewer's Responses to Questions

**Comments to the Author**

1. Is the manuscript technically sound, and do the data support the conclusions?

Reviewer #1: Partly

Reviewer #2: Yes

2. Has the statistical analysis been performed appropriately and rigorously? 

Reviewer #1: I Don't Know

Reviewer #2: Yes

3. Have the authors made all data underlying the findings in their manuscript fully available?

Reviewer #1: No

Reviewer #2: No

4. Is the manuscript presented in an intelligible fashion and written in standard English?

Reviewer #1: Yes

Reviewer #2: Yes

5. Review Comments to the Author

Reviewer #1: Plos one PONE_D_19_28331

This study uses data from 6 sites of the MAL-ED study to look at the association between stunting and cognitive scores at age 5 years.

The original study was very detailed, collecting extensive data on nutrition and illnesses as well as home background. In this context, the authors need to justify why it is useful to look at stunting – which is a result of those factors, and what their analysis adds to the papers already published using the same data

Furthermore, thought needs to be given to the likely causal pathways. Does it make sense to adjust for factors like the pathogen load, when this may be one of the causes of stunting? Adjusting for the causes of stunting risks over-adjustment.

The outcome is the WPPSI score. I understand that this has been published previously but it would still be helpful to provide more information in this paper. How were the scales adapted for cultural appropriateness? Was this different in each country? How was it validated?

I think – though it is not entirely clear from text or tables - that they used a measure of “fluid reasoning” rather than the whole WPSSI score. And that a constant scale was used across sites although the mean score was very different across the sites. Given these large differences, wouldn’t a relative within site scale be more appropriate for looking at the associations with stunting ?

And given that the proportion stunted varies so much by site, wouldn’t it be better to include site as a confounder than as a cluster variable?

There is no mention of possible interactions (by site or by other variables).

Some of the other variables also need more explaining:

What is the rationale for grouping water and sanitation, assets, maternal education and income into a composite score? Water and sanitation could have direct effects on infections and health which is different from the indirect effects of income and assets.

The HOME inventory needs more explanation

The transferrin receptor is said to be important for growth so may also be a cause of stunting. Since they have previously reported the transferrin receptor as a risk factor for cognitive development it should not be one of the results mentioned in the abstract.

For the analysis of risk factors for early stunting it is not clear what the outcome is. Is it first measure of stunting within the first 2 years? Does it make sense to analyse this whole time period together? Risk factors for stunting in the first few months are likely to be different from later onset.

Figure 2 looks odd as it implies that no child was stunted at birth. Some left censoring is needed

The analysis is based on 60% of the children enrolled. This loss to follow-up and any bias it may have introduced need to be discussed.

Other points:

Please avoid non-standard abbreviations in tables and figures

Line 158: why multiply by 365?

The conclusions call for a comparison with high income settings, but the causes of stunting in these settings are likely to be very different.

Reviewer #2: This manuscript presents an analysis of the MAL-ED LIMC multicountry cohort to explore the important association between early nutrition and later cognitive function. It largely confirms the previous findings that poor growth in early life is associated with poor cognitive development, also previously known information on timing and persistence of stunting. It controls for other contributing factors.

The writing is clear and precise and the data and findings well presented. The statistical methods used (though not my expertise) seem appropriate.

Although the findings are not particularly novel or unexpected, this is a useful addition to the literature. It strength is the multicountry approach, although given that the settings were so different, it might be helpful to explore whether there was interaction between site and any of the variables in the models and to stratify the analysis where appropriate.

In the discussion the recommendation that the analysis could help with identification of specific higher risk groups for targeting misses some important principles (about absolute versus relative risk, distribution of the risk factor (50:50!) and equity) when suggesting eg that males could be targeted for interventions. Whilst males might be at higher risk in this analysis, an intervention which excludes female infants would exclude a very high proportion of infants who would develop stunting, be inequitable, and in any case, many of the interventions might be applied before the sex of the infant is known.

6. PLOS authors have the option to publish the peer review history of their article (what does this mean?). If published, this will include your full peer review and any attached files.

Reviewer #1: No

Reviewer #2: No

---

## [Author Response · Author response to Decision Letter 0]

26 Dec 2019

We have attached "Response to Reviewers" file.

---

## [Editor Report · Decision Letter 1]

31 Dec 2019

Impact of early-onset persistent stunting on cognitive development at 5 years of age: Results from a multi-country cohort study

PONE-D-19-28331R1

Dear Dr. Mahfuz,

We are pleased to inform you that your manuscript has been judged scientifically suitable for publication and will be formally accepted for publication once it complies with all outstanding technical requirements.

With kind regards,

Samson Gebremedhin, PhD

Academic Editor

PLOS ONE

---

## [Editor Report · Acceptance letter]

15 Jan 2020

PONE-D-19-28331R1 

Impact of early-onset persistent stunting on cognitive development at 5 years of age: Results from a multi-country cohort study 

Dear Dr. Mahfuz:

I am pleased to inform you that your manuscript has been deemed suitable for publication in PLOS ONE. Congratulations! Your manuscript is now with our production department. 

With kind regards,

on behalf of

Dr. Samson Gebremedhin 

Academic Editor

PLOS ONE